# Differences in Fatty Acid Profile and Physical-Chemical Composition of *Slavonska slanina*—Dry Cured Smoked Bacon Produced from Black Slavonian Pig and Modern Pigs

**DOI:** 10.3390/ani12070924

**Published:** 2022-04-04

**Authors:** Katarina Latin, Krešimir Mastanjević, Nikola Raguž, Mateja Bulaić, Ras Lužaić, Marija Heffer, Boris Lukić

**Affiliations:** 1Black Slavonian Pig Breeders Association “Fajferica”, Vladimira Nazora 1, 31400 Đakovo, Croatia; latinnkata@gmail.com; 2Faculty of Food Technology Osijek, Josip Juraj Strossmayer University of Osijek, F. Kuhača 20, 31000 Osijek, Croatia; 3Department for Animal Production and Biotechnology, Faculty of Agrobiotechnical Sciences Osijek, Josip Juraj Strossmayer University of Osijek, Vladimira Preloga 1, 31000 Osijek, Croatia; nikola.raguz@fazos.hr (N.R.); ras.luzaic@fazos.hr (R.L.); blukic@fazos.hr (B.L.); 4Inspecto d.o.o., Vukovarska Cesta 239b, 31000 Osijek, Croatia; mateja.bulaic@inspecto.hr; 5Department of Medical Biology and Genetics, Faculty of Medicine, Josip Juraj Strossmayer University of Osijek, Josipa Huttlera 4, 31000 Osijek, Croatia; mheffer@mefos.hr

**Keywords:** fatty acid profile, dry cured bacon, physico-chemical composition, Black Slavonian Pig

## Abstract

**Simple Summary:**

Research on the products of local pig breeds is quite challenging as the populations are small, breeders are less educated and it is often difficult to design systematic research in various and usually different breeding conditions, compared to commercial breeds. In this study, we have analysed dry cured bacons, produced from the local Black Slavonian and modern breeds on the small home-made scale, and compared their physico-chemical properties and fatty acid profiles. Results have showed significant differences in chemical composition and fatty acid profiles between analysed pig breeds.

**Abstract:**

The objective of this study was to compare the psysico-chemical composition and fatty acid (FFA) profile of traditional dry cured bacon *Slavonska slanina*, produced from the authentic Black Slavonian Pig (BP) and modern pigs (MP), using traditional homemade principles. The samples of *Slavonska slanina* produced from BP had a significantly (*p* < 0.05) higher fat content (78.32%) than the samples produced from MP (46.47%), while the samples produced from MP showed significantly higher (*p* < 0.05) protein and moisture content. The samples produced from BP also showed lower *a*_w_ and salt content but higher pH. Determination of fatty acid composition was performed at the end of the production process. The composition of fatty acids with respect to the groups of saturated (SFA), monounsaturated (MUFA) and polyunsaturated (PUFA) fatty acids was determined, as well as the ratios *n*-6/*n*-3 and PUFA/SFA. The results of fatty acid composition determination of *Slavonska slanina* produced from BP and MP show that oleic acid (C18: 1n9) is the most dominant fatty acid from the MUFA group (47.02 and 46.25%), the most common SFA acid was palmitic acid (C16: 0) (23.44 and 24.96%), and PUFA linoleic acid (C18: 2*n*-6c) (10.76 and 9.74%). The genotype had a significant (*p* < 0.05) effect on the SFA and USFA composition of *Slavonska slanina.* The ratios PUFA/SFA (0.34–0.28) and *n*-6/*n*-3 (31.84–27.34) for samples of *Slavonska slanina* produced from BP and MP are in concordance with previously published data for different dry cured meat products, and do not comply with the nutritionally recommended values of international health organizations (PUFA/SFA > 0.4 and *n*-6/*n*-3 < 4).

## 1. Introduction

Slavonia (Eastern Croatia) is a region known for the production of traditional dry cured products from pigs, dating back to the time of family cooperatives and the existence of farms located on pastures near forests, where pigs and other livestock were raised.

The traditional dry cured products in Slavonia are generally produced from the meat of modern pig breeds such as Landrace or Large White. Nowadays, when the market trends shift towards sustainable animal production with local autochthonous breeds, the Croatian Black Slavonian pig becomes very popular with notable population growth [1]. As the quality of these products such as home-made ham, dry cured bacon and various types of dry sausages is expected to be higher for consumers, products of Black Slavonian pig slowly become recognizable brand in the domestic and international market [2].

The quality of these products is highly important to investigate, especially their fat content and its profiles, because the high fat content or even their unbalanced composition [3,4] leads to various chronic cardiovascular diseases [5,6].

*Slavonska slanina* is dry cured bacon produced from the pork belly with fat, muscle and skin tissue which is preserved by salting, smoking, drying and ripening. The bacon production process begins with the selection and processing of pork belly followed by the salting process, which lasts approximately 10 days. Smoking usually lasts from 7 to 10 days, which is followed by drying and ripening [7]. Due to the various technological processes during the production of traditional dry cured products, lipids undergo a series of transformations that include hydrolytic processes, release of short-chain fatty acids and oxidation, with the formation of peroxides and volatile components, thus contributing to the aroma of the final product [8].

Lipids of dry cured bacon such as *Slavonska slanina,* consist of glycerides composed of mono-, di-, triglycerides that are esterified to one, two or three fatty acids, as well as phospholipids and cholesterol. The amount of fatty acids is 40–50%, and the most common fatty acid is oleic, both in muscle and adipose tissue. The breakdown of muscle tissue lipids depends on the hydrolysis of the most important triglycerides, hydrolysis of phospholipids, and finally lipolysis. The lipolysis in dry cured products mainly depends on its type, the type of adipose tissue (subcutaneous, inter-, intramuscular) and the amount of endogenous lipolytic enzymes [9,10].

The most common fatty acids in dry cured meat products belong to MUFA (41–59%), followed by SFA (30–45%) and PUFA (9–18%) [11,12,13]. In the MUFA group, oleic (C18: 1n9) (38–42%) and palmitoleic (C16: 1n7) (2–3%) fatty acids are the most common [9]. The main SFAs are palmitic (C16: 0) (23–24%) and stearic (C18: 0) (10–15%) acid. The main component of PUFA is linoleic acid (C18: 2*n*-6) with a content of 6–16%, which is generally lower in cured meat products (7–10%) compared to fermented sausages (10–16%). The highest presence of oleic, palmitic, linoleic and stearic fatty acids in mature dry cured product (prosciutto, dry ham) is the result of their highest initial concentration in fresh meat and their stability and resistance to oxidation [14]. The formation of fatty acids and their decomposition into short-chain fatty acids and oxidation are the most important reactions during the maturation process that affects the formation of a specific taste and smell. Lipolysis of products plays a significant role in creating the aroma and flavour components of dry cured meat product and their precursors [15,16].

Nevertheless, the fatty acid composition of meat and adipose tissues depends on the various and numerous factors. These factors include genetics/breed, nutrition, farm conditions, final weight, age, sex, carcass composition etc. [17]. Breed is shown to be a strong effect on the fatty acid composition in muscle and fat tissue [3,18,19].

In the available scientific literature and research, the studies about the quality and fatty acid composition of the fresh meat and fat tissue predominate. If the quality and fatty acid profiles are analysed on the products, focus is mainly given to the popular traditional products such as ham or prosciutto, whereas dry cured bacon is generally circumvented. In addition, research on the products of local breeds is quite challenging as the populations are small, breeders are less educated and it is often difficult to design systematic research in various and usually different breeding conditions, compared to commercial breeds. In this study, we were guided by the idea to analyse products from the small and home-made producers, which people in Eastern Croatia traditionally consume, therefore our aim was to determine and compare the physico-chemical properties and fatty acid profiles of dry cured bacon *Slavonska slanina*, made from Black Slavonian and modern pig breeds, and finally, to address if the local breeds provide higher quality products.

## 2. Materials and Methods

### 2.1. Animals Used for Slavonska slanina Production

All animals were raised on small family farms, traditional for Slavonian region. Samples of dry cured bacon made from the 29 barrows of Black Slavonian Pig (BP) and 30 from the modern pigs (MP; Landrace, Large white and their crosses) were collected from the 59 local pig producers who traditionally produce dry cured bacon. All farms were indoor type with small outdoor space available, which allowed pigs to freely walk outside the barn. Feeding of both groups was the same, with feeding mixture composed of corn (50%), barley (25%), oat (25%) and fresh green clover given upon availability. Average slaughter weight of animals for both groups was 180 ± 20 kg.

### 2.2. Production of Slavonska slanina

The traditional production of the *Slavonska slanina* starts at the end of November and lasts until the late April. Dry cured bacon samples (weighing cca. 8 kg) made from BP and MP were collected from the local producers originated from *Slavonia* region at the beginning of April. Technological procedures were the same for all produced bacons, which includes shaping, trimming and rib bones removal in traditional Slavonian manner. The raw bellies were firstly dry-salted for three weeks with unlimited amount of salt. Thereafter, the bellies were rinsed with tap water to remove the excess salt. For the smoke production the hard wood (hornbeam and beech) was used. The bacons were exposed to the smoke every other day, for 3–4 h, for a total of 14–21 days. This means that the bacon was exposed to smoke for approximately 7 times, while the total production of *Slavonska slanina* lasted approximately 90 days.

Temperature and relative humidity, during the smoking, drying and ripening period varied depending on natural climate conditions in Eastern Croatia. The temperature for smoking stage varied between 2.5 and 12.4 °C and the relative humidity from 64.1 to 94.7%. During the drying and ripening period (45 days) temperature varied from 2 to 15 °C and the relative humidity from 60.3 to 92.8%.

### 2.3. Determination of Physical-Chemical Properties

Mass fraction of protein, fat moisture and collagen in the samples of *Slavonska slanina* were determined by Foss Food Scan Meat Analyzer according to the AOAC method April 2007 [20].

pH/Ion-Bench pH/Ion/mV meter (Eutech Instruments Pte Ltd., Singapore) was used to determine the samples pH value in ordinance with the ISO 2917: 1999.

HygroLab 3-Multi-channel Humidity & Water Activity Analyzer (Rotronic, Bassersdorf, Switzerland) was used for determination of water activity (*a*_w_). Salt (sodium chloride) content was determined according to the ISO method 1841.

Three determinations for general composition, pH, *a*_w_ and salt content were measured from each sample.

### 2.4. Determination of Fatty Acids Profile

Lipids obtained after Soxhlet extraction of samples were converted to corresponding FAMEs by trans-esterification with potassium hydroxide (ISO 12966-2). The whole bacon was homogenized from which the three samples were taken. Approximately 100 mg of sample was dissolved in 2 mL of isooctane in a test tube and 100 µL of methanolic potassium hydroxide solution (2 mol/L) was added. Solution was shaken vigorously for about 60 s. Neutralization of solution was conducted by addition of 1 g of sodium hydrogen sulfate, anhydrous. After the salt has settled, 1 mL of upper phase was transferred into 2 mL vial and analysed.

GC analyses were performed on Agilent 7890B equipped with autosampler, flame ionization detector and split/splitless injector together with Agilent DB-23 GC column, with 60 m length, I.D. 0.25mm, film thickness 0.25 µm. Injector temperature was set at 250 °C to which 1 µL of sample was introduced at a split ratio of 1:10.

The oven temperature was set at 60 °C and rising to 220 °C at 7 °C/min with hold of 20 min. Helium was used as the carrier gas at an isobaric pressure of 40.0 psi. Detector temperature was set at 250 °C with N_2_ „makeup“ flow of 25 mL/min, H_2_ 30 mL/min and air 400 mL/min flow. Agilent Openlab Chemstation chromatographic software was used for data collection and calculation of all parameters.

### 2.5. Statistical Analyses

Statistical analysis of the data was performed in Statistica 13.1. software (TIBCO Software Inc., Palo Alto, CA, USA). The values of all traits between two investigated groups of BP and MP were analysed using *t*-test.

## 3. Results and Discussion

### 3.1. Physical-Chemical Properties

Physical-chemical properties of *Slavonska slanina* produced from MP and BP are presented in Table 1. The samples originated from BP showed higher fat content (78.32%) than the samples from MP (46.47%) with statistical significance (*p* < 0.05). This is probably related to the nature of the samples (BP) whose fresh meat has a higher amount of fat in carcasses and higher intermuscular fat content [2]. On the other hand, samples produced from MP showed significant (*p* < 0.05) higher protein (16.67%) and moisture content (32.72%).

The bacons from BP had lower salt content (3.33%) which is probably related to the higher fat content in the samples. The salt must diffuse through the liquid phase in muscle, so the salt diffusion through the fat and skin is negligible [9]. pH was higher in bacons from BP than in bacons from MP but without statistical significance observed (*p* > 0.05).

The BP bacons showed lower *a*_w_ (0.70) than the samples produced from MP (0.87). For the Croatian Pancetta, which is a similar dry cured product produced from modern pigs, Pleadin et al. (2021) [21] reported similar salt content (4.51%), higher fat (47.86%) and protein content (20.58%), but lower moisture content (23.24%) compared to bacon made from MP. This can be related to the longer drying period of Croatian Pancetta. The bacons from BP showed large differences in all physico-chemical properties than the bacons made from MP and also from Croatian Pancetta.

### 3.2. Fatty Acids Profile

The fatty acid composition of *Slavonska slanina* after 90 days of production is shown in Table 2. Dry cured meat products from modern pig breeds contain on average 35–40% saturated fatty acids (SFA), 45–50% monounsaturated fatty acids (MUFA) and 10–15% of polyunsaturated fatty acids (PUFA) [10]. *Slavonska slanina* produced from MP contained 39.20% SFA, 49.91% MUFA and 10.86% PUFA, and *Slavonska slanina* from BP 36.14% SFA, 51.68% MUFA and 11.96% PUFA. Samples of *Slavonska slanina* produced from MP had significantly (*p* < 0.05) higher shares of SFA, compared to samples produced from BP (Table 2). The bacon made from BP had higher content of MUFA and PUFA than the samples produced from MP but without significance (*p* > 0.05). The higher content of MUFA and PUFA in samples produced from BP are probably related to different genotype.

Similar dry cured product as *Slavonska slanina* Croatian Pancetta had a higher SFA content (43.13%), lower MUFA content (44.66%) but higher PUFA content (12.14%) [21]. On the other hand, Bayonne ham (36.52%) [22], Jinhua ham (37.10%) [23], Serrano ham (36.71%) [10] and Vipava prosciutto (38%) [24] have a similar share of SFA as *Slavonska slanina*. Dalmatian prosciutto had a higher share of SFA (41.82%) [12,25], Istrian prosciutto (39.21–42.77%) [26], Drniš prosciutto (45.19%) [27] and Kraški prosciutto (57.2–59.3%) [25]. Parma prosciutto (35.58%) [28], Cinta Sinese (33.26%) and Tuscan prosciutto (32.30–34.20%) [29] had a lower share of SFA.

MUFA content in Iberian prosciutto (up to 60%) [30] is the result of special genotype of Iberian pigs, where at least the last two months of fattening before slaughtering pigs were fed on pasture with acorns. MUFA content in *Slavonska slanina* from BP was 51.58% which is higher than the MUFA content in samples from MP (49.91%). Both groups showed really high MUFA content (Table 2).

Literature data show that oleic fatty acid is the most abundant in traditional dry cured meat products, followed by palmitic (C16: 0), stearic (C18: 0) and linoleic acid [12,21,26,31,32].

In general, monounsaturated fatty acids (MUFAs) predominate in dry cured meat products, while polyunsaturated fatty acids (PUFAs) are the least common.

Oleic acid (C18: 1*n*-9c) from the MUFA group, with a mass fraction of 47.02% in samples from BP and 46.25% in samples from MP was the most dominant fatty acid in both groups of *Slavonska slanina*. Compared to the mass fraction of oleic acid in similar dry cured meat products, *Slavonska slanina* had similar mass fractions of oleic acid as Bayonne ham [12] and Kraški prosciutto [25]. Higher levels of oleic acid were determined in Iberian and Serrano prosciutto [11], Dalmatian and Istrian prosciutto [26] and in Parma prosciutto [33], which may be associated with a longer ripening time, but also with the genotype and pig diets.

From the SFA group, palmitic acid was the most represented (C16: 0) (24.96% for samples from MP and 23.44% for samples from BP). Linoleic acid (C18: 2*n*-6c) had the largest share from the PUFA group for samples produced from MP 9.74% and 10.76% for samples produced from BP (Table 2).

The type of feeding, regime and the composition of feed given to the animals, decisively influences the fatty acid composition of intramuscular fat. Fatty acids from feed are converted into the adipose tissue of pigs, and the degree of incorporation depends on the specifics of fatty acids and the type of feed.

The samples produced from BP showed large variation in fatty acid composition, especially in SFA, USFA, *n*-6, *n*-3, MUFA/SFA AND PUFA/SFA (Figure 1). This can be related to the high variation in BP genotype, which is finally as expected due to the current practices in breeding program of Black Slavonian pigs [34].

Nutritionally desirable properties of fatty acids of *Slavonska slanina* produced from BP and MP are generally determined through the ratio of PUFA/SFA and *n*-6/*n*-3 (Table 2). PUFA/SFA ratios above 0.4 and *n*-6/*n*-3 below 4 are considered as recommendable [3,17]. In this study, the PUFA/SFA ratio in samples from BP was 0.34, while for MP was 0.28. *n*-6/*n*-3 ratio in this study for the MP bacons was 27.5, and 31.84 for BP bacons. The results confirm that, given the above recommendations, *Slavonska slanina* made from both genotypes is not within the recommended values for the PUFA/SFA ratio, as well as for the *n*-6/*n*-3 ratio.

## 4. Conclusions

*Slavonska slanina* produced from BP had a higher fat content and pH value, but lower protein, moisture, salt content and *a*_w_. The products from BP showed significantly lower SFA and significantly higher USFA values, but also the higher omega 3 and omega 6 fatty acid values which implies to the different BP characteristics, which is expected as Black Slavonian breed is local breed without selection implemented towards higher leanness, lower fat thickness and generally improved production traits.

Overall, higher variation is observed for the majority of the chemical and fatty acid profile properties for the BP group, which seems as a logical reflection of a breeding practice, without precise and systematic breeding goals such as carcass and meat quality traits.

These results could help in breeding program of BP by identifying individuals with phenotypes possessing nutritionally desirable properties and therefore utilized in future selection practices.

## Figures and Tables

**Figure 1 animals-12-00924-f001:**
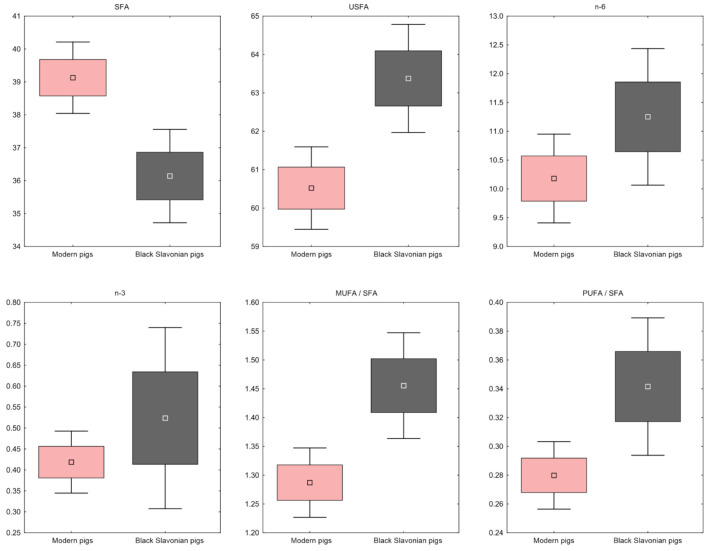
Box plots showing the differences in main fatty acids; SFA, USFA, *n*-6, *n*-3 and ratios MUFA/SFA and PUFA/SFA between the BP and MP groups. The differences between groups were statistically significant for SFA, USFA, MUFA/SFA and PUFA/SFA as *p* < 0.05. Inner points within boxes represent mean, boxes represent mean ± SE, and whiskers represent mean ± 1.96*SE.

**Table 1 animals-12-00924-t001:** Physical-Chemical Properties of *Slavonska slanina*.

	MP	BP	*p*-Value
Fat (%)	46.47 ^b^ ± 0.75	78.32 ^a^ ± 0.80	<0.05
Protein (%)	16.64 ^a^ ± 0.21	7.66 ^b^ ± 0.25	<0.05
Moisture (%)	32.72 ^b^ ± 0.63	12.91 ^a^ ± 0.91	<0.05
Collagen (%)	1.29 ^a^ ± 0.14	2.86 ^a^ ± 0,21	ns
Salt (%)	4.18 ^a^ ± 0.22	3.33 ^b^ ± 0,41	<0.05
pH	5.59 ^a^ ± 0.52	5.69 ^a^ ± 0.33	ns
*a* _w_	0.87 ^a^ ± 0.45	0.70 ^b^ ± 0.39	<0.05

^a,b^ Values in a row (± standard deviation) with different letters are significantly different (*p* < 0.05) (*t*-test); equal letters for groups mean no differences for that characteristic; ns = not significant; BP (Black Slavonian Pig), MP (Modern Breeds).

**Table 2 animals-12-00924-t002:** Fatty (mass fraction, %) Acids Profile of *Slavonska slanina*.

	MP	BP	*p*-Value
C10:0	0.07 ^a^ ± 0.02	0.04 ^b^ ± 0.03	<0.05
C12:0	0.07 ^a^ ± 0.01	0.05 ^b^ ± 0.03	<0.05
C14:0	1.35 ± 0.15	1.37 ± 0.22	ns
C15:0	0.02 ± 0.03	0.02 ± 0.03	ns
C16:0	24.96 ^a^ ± 1.56	23.44 ^b^ ± 2.29	<0.05
C16:1	2.36 ^b^ ± 0.39	2.81 ^a^ ± 0.72	<0.05
C17:0	0.30 ± 0.13	0.25 ± 0.18	ns
C17:1	0.26 ^a^ ± 0.13	0.17 ^b^ ± 0.17	<0.05
C18:0	12.25 ^a^ ± 1.60	10.72 ^b^ ± 2.12	<0.05
C18:1*n*-9t	0.17 ^b^ ± 0.16	0.53 ^a^ ± 0.73	<0.05
C18:1*n*-9c	46.25 ± 3.12	47.02 ± 3.55	ns
C18:2*n*-6t	0.09 ± 0.02	0.15 ± 0.44	ns
C18:2*n*-6	9.74 ± 2.02	10.76 ± 3.13	ns
C20:1	0.87 ^b^ ± 0.16	1.15 ^a^ ± 0.26	<0.001
C20:0	0.18 ^a^ ± 0.04	0.12 ^b^ ± 0.01	<0.05
C18:3*n*-3	0.39 ± 0.18	0.37 ± 0.44	ns
C20:2:*n*-6	0.44 ± 0.15	0.48 ± 0.27	ns
C20:3:*n*-6	0.02 ± 0.02	0.03 ± 0.04	ns
C20:3:*n*-3	0.03 ± 0.03	0.05 ± 0.08	ns
C20:4:*n*-6	0.15 ± 0.04	0.11 ± 0.10	ns
∑ SFA	39.20 ^a^ ± 2.97	36.14 ^b^ ± 3.96	<0.05
∑ USFA	60.77 ^b^ ± 2.95	63.38 ^a^ ± 3.94	<0.05
∑ MUFA	49.91 ± 3.32	51.68 ± 3.71	ns
∑ PUFA	10.86 ± 2.27	11.96 ± 3.36	ns
∑ PUFA/∑SFA	0.28 ^b^ ± 0.06	0.34 ^a^ ± 0.13	<0.05
∑MUFA/∑SFA	1.27 ^b^ ± 0.16	1.46 ^a^ ± 0.26	<0.05
∑ *n*-3	0.42 ± 0.20	0.52 ± 0.54	ns
∑ *n*-6	10.18 ± 2.12	11.25 ± 3.31	ns
∑ *n*-3/∑ *n*-6	27.54 ± 8.65	31.84 ± 23.02	ns

^a,b^ Values in a row (± standard deviation) with different letters are significantly different (*p* < 0.05) (*t*-test); equal letters for groups mean no differences for that characteristic; ns = not significant; BP (Black Slavonian Pig), MP (Modern Pigs).

## Data Availability

The data presented in this study are available on request from the corresponding author. The data are not publicly available due it is a property of Black Slavonian Pig Breeders association “Fajferica”.

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
