# Peer review of "Differences in Fatty Acid Profile and Physical-Chemical Composition of Slavonska slanina—Dry Cured Smoked Bacon Produced from Black Slavonian Pig and Modern Pigs"

_animals, 2022, doi:10.3390/ani12070924_

Round 1

Reviewer 1 Report

This is a well written paper showing an interesting comparison between Black Slavonian and modern pigs.  It can be used as sort of a benchmark showing how products from these lines have changed over time and how products made from them may have changed as well.  

A few comments:

The authors do not have to include the word "statistically" each time they want to denote a difference between two mean values when the also include the P values.  The statistical part is assumed.

Please have a second look at the Keywords proposed in the paper.  

Two sentences in the introduction seems awkwardly written.  Please look on line 53 and line 56.

The sentence regarding the fatty acid profiles not fitting the recommendations seems a bit out of place.  Not sure how to remedy this.  Perhaps a bit more discussion about it or just leave it alone.  It seems like a bit of an after thought as written.  But I guess it follows from the all the time and expense of conducting the fatty acid profile.

Author Response

This is a well written paper showing an interesting comparison between Black Slavonian and modern pigs.  It can be used as sort of a benchmark showing how products from these lines have changed over time and how products made from them may have changed as well. 

Thanks to the reviewer for the nice recommendation.

A few comments:

The authors do not have to include the word "statistically" each time they want to denote a difference between two mean values when the also include the P values.  The statistical part is assumed.

Yes, we agree to the reviewer, this is corrected.

Please have a second look at the Keywords proposed in the paper. 

Yes, we agree to the reviewer, this is corrected.

Two sentences in the introduction seems awkwardly written.  Please look on line 53 and line 56.

Yes, we agree to the reviewer, this is corrected.

The sentence regarding the fatty acid profiles not fitting the recommendations seems a bit out of place.  Not sure how to remedy this.  Perhaps a bit more discussion about it or just leave it alone.  It seems like a bit of an after thought as written.  But I guess it follows from the all the time and expense of conducting the fatty acid profile.

We agree with the reviewer, the nutritional recommendation for this kind of meat product are slightly out of scope, but the authors agreed to leave this sentence. In our bigger ongoing research we will try to address this point!

This is a well written paper showing an interesting comparison between Black Slavonian and modern pigs.  It can be used as sort of a benchmark showing how products from these lines have changed over time and how products made from them may have changed as well. 

Thanks to the reviewer for the nice recommendation.

A few comments:

The authors do not have to include the word "statistically" each time they want to denote a difference between two mean values when the also include the P values.  The statistical part is assumed.

Yes, we agree to the reviewer, this is corrected.

Please have a second look at the Keywords proposed in the paper. 

Yes, we agree to the reviewer, this is corrected.

Two sentences in the introduction seems awkwardly written.  Please look on line 53 and line 56.

Yes, we agree to the reviewer, this is corrected.

The sentence regarding the fatty acid profiles not fitting the recommendations seems a bit out of place.  Not sure how to remedy this.  Perhaps a bit more discussion about it or just leave it alone.  It seems like a bit of an after thought as written.  But I guess it follows from the all the time and expense of conducting the fatty acid profile.

We agree with the reviewer, the nutritional recommendation for this kind of meat product are slightly out of scope, but the authors agreed to leave this sentence. In our bigger ongoing research we will try to address this point!

Reviewer 2 Report

The text submitted for review may be accepted for publication as a communication, after taking into account the corrections. Materials and Methods planned in these studies do not meet the assumptions of the research experience, but the collected data can be disseminated as a short communication. However, the manuscript needs to be improved. The chapter "Conclusions" has to be changed as it is an abbreviated repetition of the results. Please, prepare 2-3 sentences summarizing the results obtained and indicating the possible application/usefulness of these results in practice. Delete the Figure 1, as it is a repetition of the data presented in table 2.

Author Response

The text submitted for review may be accepted for publication as a communication, after taking into account the corrections. Materials and Methods planned in these studies do not meet the assumptions of the research experience, but the collected data can be disseminated as a short communication.

However, the manuscript needs to be improved. The chapter "Conclusions" has to be changed as it is an abbreviated repetition of the results.

Thanks to the reviewer for his comment and recommendations, we have addressed his points.

Please, prepare 2-3 sentences summarizing the results obtained and indicating the possible application/usefulness of these results in practice.

Yes, we agree to the reviewer, this is corrected. These results could be included in future breeding program by identifying desirable phenotypes and used for selection of elite parents.

Delete the Figure 1, as it is a repetition of the data presented in table 2.

Yes, we generally agree with the reviewers point. However, this paper is submitted as short communication and the purpose of Figure 1 is to give the better insight into the large variation in fatty acid profile of BP. As described in the paper, one of the conclusions is beside that BP had higher some of the important phenotypes, it also showed massive within variation compared to the MP (which is nicely visible in the figure), therefore we left the Figure 1. It is always more intuitive to present results visually instead of plain numbers in tables, therefore we finally decided to leave the both Figure 1 and Tables 1 and 2.

Reviewer 3 Report

Generally a well written manuscript. Some changes, clarification are required. I made my comment in the PDF file, I upload it.

Author Response

In this way it means to me that PUFA content has to be more than 40 % of SFA. For me it seems to be a high level, of which may cause problems regarding rancidity and product texture. Double check the value and unit.

Thanks to the reviewer for his comments and recommendations for paper improvement.

This recommendation is given by the recognised international health organisations (Wood, 2004), while the reference in only aimed to nutritional and health aspects of human nutrition. People should be aware of that while consuming these kind of products, which is the reason why we think it is better to provide the given recommendations.

I think similar trend is true for other countries, where fatty pig breeds are exist.

We agree with the reviewer’s comment, we corrected this.

what kind of feeding leads to this PUFA content.

We agree with the reviewer’s comment, it looks wide at the first glance. The PUFA content referred in this section is a result of various genotypes, feeding and technological production factors. That’s why is it so wide.

What kind of clover?

In Croatia, breeders usually use fresh clover (white and red) as pig feed, which are similar in chemical composition, therefore we didn’t describe which one was used, as it varies depending on the various factors (vegetation, previous culture, region etc.).

Describe sampling methods.

We agree with the reviewer’s comment, we corrected this.

The numerical difference is so small, that it is not important even from professional point of view. Therefore, I did not mention it.

We agree with the reviewer’s comment, we corrected this.

if you use superscripts, than the p-values in that way are meaningless. I would delete this column.

Yes, this is true. However, we used two different significance levels (0.01 and 0.05), and as the table is quite simple in representation (only three columns), we decided to leave the p values, then the readers could identify differences simpler and faster.

This needs to be explained under the table

This is a symbol for water activity, it is showed earlier at the beginning of the manuscript.

What are the units of values? What values are of the second ones (sd or se)? true for other tables as well! These are not "free" fatty acids.  Tables should appear after the first mention. This is not true for table 2,

We agree with the reviewer’s comment. This was corrected.

In my opinion this term is not good here. Free fatty acids means that they are not bound to glycerol. But this is not true here. It would need different analyses. Correct it. It is simple fatty acid profile.

We agree with the reviewer’s comment. This was corrected throughout whole paper.

The same data has to be presented always only once, either table or figure, choose.

Yes, we generally agree with the reviewers point. However, this paper is submitted as short communication and the purpose of Figure 1 is to give the better insight into the large variation in fatty acid profile of BP. As described in the paper, one of the conclusions is beside that BP had higher some of the important phenotypes, it also showed massive within variation compared to the MP (which is nicely visible in the figure), therefore we left the Figure 1. It is always more intuitive to present results visually instead of plain numbers in tables, therefore we finally decided to leave the both Figure 1 and Tables 1 and 2 (if someone wish to see the exact differences in some of the acids).